# Visual Reinforcement Learning with Imagined Goals

**Ashvin Nair**[*]**, Vitchyr Pong**[*]**, Murtaza Dalal, Shikhar Bahl, Steven Lin, Sergey Levine**
University of California, Berkeley
{anair17,vitchyr,mdalal,shikharbahl,stevenlin598,svlevine}@berkeley.edu

## Abstract

For an autonomous agent to fulfill a wide range of user-specified goals at test time, it must be able to learn broadly applicable and general-purpose skill repertoires. Furthermore, to provide the requisite level of generality, these skills must handle raw sensory input such as images. In this paper, we propose an algorithm that acquires such general-purpose skills by combining unsupervised representation learning and reinforcement learning of goal-conditioned policies. Since the particular goals that might be required at test-time are not known in advance, the agent performs a self-supervised "practice" phase where it imagines goals and attempts to achieve them. We learn a visual representation with three distinct purposes: sampling goals for self-supervised practice, providing a structured transformation of raw sensory inputs, and computing a reward signal for goal reaching. We also propose a retroactive goal relabeling scheme to further improve the sample-efficiency of our method. Our off-policy algorithm is efficient enough to learn policies that operate on raw image observations and goals for a real-world robotic system, and substantially outperforms prior techniques.

## 1 Introduction

Reinforcement learning (RL) algorithms hold the promise of allowing autonomous agents, such as robots, to learn to accomplish arbitrary tasks. However, the standard RL framework involves learning policies that are specific to individual tasks, which are defined by hand-specified reward functions. Agents that exist persistently in the world can prepare to solve diverse tasks by setting their own goals, practicing complex behaviors, and learning about the world around them. In fact, humans are very proficient at setting abstract goals for themselves, and evidence shows that this behavior is already present from early infancy [43], albeit with simple goals such as reaching. The behavior and representation of goals grows more complex over time as they learn how to manipulate objects and locomote. How can we begin to devise a reinforcement learning system that sets its own goals and learns from experience with minimal outside intervention and manual engineering?

In this paper, we take a step toward this goal by designing an RL framework that jointly learns representations of raw sensory inputs and policies that achieve arbitrary goals under this representation by practicing to reach self-specified random goals during training. To provide for automated and flexible goal-setting, we must first choose how a general goal can be specified for an agent interacting with a complex and highly variable environment. Even providing the state of such an environment to a policy is a challenge. For instance, a task that requires a robot to manipulate various objects would require a combinatorial representation, reflecting variability in the number and type of objects in the current scene. Directly using raw sensory signals, such as images, avoids this challenge, but learning from raw images is substantially harder. In particular, pixel-wise Euclidean distance is not an effective reward function for visual tasks since distances between images do not correspond to meaningful distances between states [36, 49]. Furthermore, although end-to-end model-free

---

[*]Equal contribution. Order was determined by coin flip.

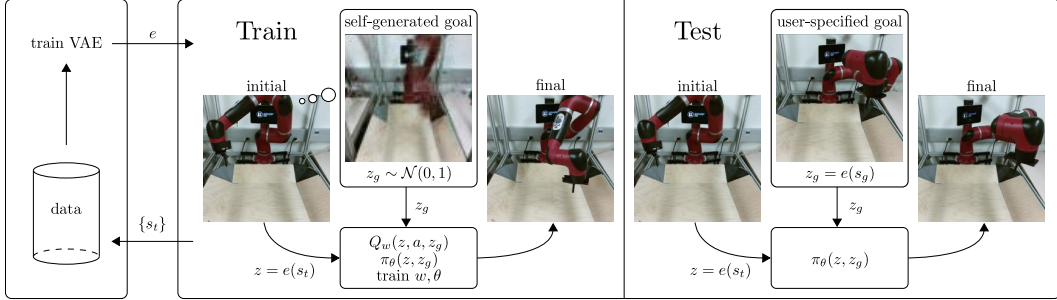

Figure 1: We train a VAE using data generated by our exploration policy (left). We use the VAE for multiple purposes during training time (middle): to sample goals to train the policy, to embed the observations into a latent space, and to compute distances in the latent space. During test time (right), we embed a specified goal observation $o_g$ into a goal latent $z_g$ as input to the policy. Videos of our method can be found at `sites.google.com/site/visualrlwithimaginedgoals`

reinforcement learning can handle image observations, this comes at a high cost in sample complexity, making it difficult to use in the real world.

We propose to address both challenges by incorporating unsupervised representation learning into goal-conditioned policies. In our method, which is illustrated in Figure 1, a representation of raw sensory inputs is learned by means of a latent variable model, which in our case is based on the variational autoencoder (VAE) [19]. This model serves three complementary purposes. First, it provides a more structured representation of sensory inputs for RL, making it feasible to learn from images even in the real world. Second, it allows for sampling of new states, which can be used to set synthetic goals during training to allow the goal-conditioned policy to practice diverse behaviors. We can also more efficiently utilize samples from the environment by relabeling synthetic goals in an off-policy RL algorithm, which makes our algorithm substantially more efficient. Third, the learned representation provides a space where distances are more meaningful than the original space of observations, and can therefore provide well-shaped reward functions for RL. By learning to reach random goals sampled from the latent variable model, the goal-conditioned policy learns about the world and can be used to achieve new, user-specified goals at test-time.

The main contribution of our work is a framework for learning general-purpose goal-conditioned policies that can achieve goals specified with target observations. We call our method reinforcement learning with imagined goals (RIG). RIG combines sample-efficient off-policy goal-conditioned reinforcement learning with unsupervised representation learning. We use representation learning to acquire a latent distribution that can be used to sample goals for unsupervised practice and data augmentation, to provide a well-shaped distance function for reinforcement learning, and to provide a more structured representation for the value function and policy. While several prior methods, discussed in the following section, have sought to learn goal-conditioned policies, we can do so with image goals and observations without a manually specified reward signal. Our experimental evaluation illustrates that our method substantially improves the performance of image-based reinforcement learning, can effectively learn policies for complex image-based tasks, and can be used to learn real-world robotic manipulation skills with raw image inputs. Videos of our method in simulated and real-world environments can be found at `https://sites.google.com/site/visualrlwithimaginedgoals/`.

## 2 Related Work

While prior works on vision-based deep reinforcement learning for robotics can efficiently learn a variety of behaviors such as grasping [33, 32, 24], pushing [1, 8, 10], navigation [30, 21], and other manipulation tasks [26, 23, 30], they each make assumptions that limit their applicability to training general-purpose robots. Levine et al. [23] uses time-varying models, which requires an episodic setup that makes them difficult to extend to non-episodic and continual learning scenarios. Pinto et al. [33] proposed a similar approach that uses goal images, but requires instrumented training in simulation. Lillicrap et al. [26] uses fully model-free training, but does not learn goal-conditioned skills. As we show in our experiments, this approach is very difficult to extend to the goal-conditioned setting

with image inputs. Model-based methods that predict images [48, 10, 8, 28] or learn inverse models [1] can also accommodate various goals, but tend to limit the horizon length due to model drift. To our knowledge, no prior method uses model-free RL to learn policies conditioned on a single goal image with sufficient efficiency to train directly on real-world robotic systems, without access to ground-truth state or reward information during training.

Our method uses a goal-conditioned value function [40] in order to solve more general tasks [45, 18]. To improve the sample-efficiency of our method during off-policy training, we retroactively relabel samples in the replay buffer with goals sampled from the latent representation. Goal relabeling has been explored in prior work [18, 2, 37, 25, 35]. Andrychowicz et al. [2] and Levy et al. [25] use goal relabeling for sparse rewards problems with known goal spaces, restricting the resampled goals to states encountered along that trajectory, since almost any other goal will have no reward signal. We sample random goals from our learned latent space to use as replay goals for off-policy Q-learning rather than restricting ourselves to states seen along the sampled trajectory, enabling substantially more efficient learning. We use the same goal sampling mechanism for exploration in RL. Goal setting for policy learning has previously been discussed [3] and recently Péré et al. [31] have also proposed using unsupervised learning for setting goals for exploration. However, we use a model-free Q-learning method that operates on raw state observations and actions, allowing us to solve visually and dynamically complex tasks.

A number of prior works have used unsupervised learning to acquire better representations for RL. These methods use the learned representation as a substitute for the state for the policy, but require additional information, such as access to the ground truth reward function based on the true state during training time [16, 14, 48, 11, 21, 17], expert trajectories [44], human demonstrations [42], or pre-trained object-detection features [22]. In contrast, we learn to generate goals and use the learned representation to obtain a reward function for those goals without any of these extra sources of supervision. Finn et al. [11] combine unsupervised representation learning with reinforcement learning, but in a framework that trains a policy to reach a single goal. Many prior works have also focused on learning controllable and disentangled representations [41, 5, 6, 38, 7, 46]. We use a method based on variational autoencoders, but these prior techniques are complementary to ours and could be incorporated into our method.

## 3 Background

Our method combines reinforcement learning with goal-conditioned value functions and unsupervised representation learning. Here, we briefly review the techniques that we build on in our method.

**Goal-conditioned reinforcement learning.** In reinforcement learning, the goal is to learn a policy $\pi(s_t) = a_t$ that maximizes expected return, which we denote as $R_t = \mathbb{E}[\sum_{i=t}^{T} \gamma^{(i-t)} r_i]$, where $r_i = r(s_i, a_i, s_{i+1})$ and the expectation is under the current policy and environment dynamics. Here, $s \in \mathcal{S}$ is a state observation, $a \in \mathcal{A}$ is an action, and $\gamma$ is a discount factor. Standard model-free RL learns policies that achieve a single task. If our aim is instead to obtain a policy that can accomplish a variety of tasks, we can construct a goal-conditioned policy and reward, and optimize the expected return with respect to a goal distribution: $\mathbb{E}_{g \sim G}[\mathbb{E}_{r_i, s_i \sim E, a_i \sim \pi}[R_0]]$, where $\mathcal{G}$ is the set of goals and the reward is also a function of $g$. A variety of algorithms can learn goal-conditioned policies, but to enable sample-efficient learning, we focus on algorithms that acquire goal-conditioned Q-functions, which can be trained off-policy. A goal-conditioned Q-function $Q(s, a, g)$ learns the expected return for the goal $g$ starting from state $s$ and taking action $a$. Given a state $s$, action $a$, next state $s'$, goal $g$, and correspond reward $r$, one can train an approximate Q-function parameterized by $w$ by minimizing the following Bellman error

$$\mathcal{E}(w) = \frac{1}{2} ||Q_w(s, a, g) - (r + \gamma \max_{a'} Q_{\bar{w}}(s', a', g))||^2 \tag{1}$$

where $\bar{w}$ indicates that $\bar{w}$ is treated as a constant. Crucially, one can optimize this loss using off-policy data $(s, a, s', g, r)$ with a standard actor-critic algorithm [26, 13, 27].

**Variational Autoencoders.** Variational autoencoders (VAEs) have been demonstrated to learn structured latent representations of high dimensional data [19]. The VAE consists of an encoder $q_\phi$, which maps states to latent distributions, and a decoder $p_\psi$, which maps latents to distributions over states. The encoder and decoder parameters, $\phi$ and $\psi$ respectively, are jointly trained to maximize

$$\mathcal{L}(\psi, \phi; s^{(i)}) = -\beta D_{KL}(q_\phi(z|s^{(i)})||p(z)) + \mathbb{E}_{q_\phi(z|s^{(i)})}[\log p_\psi(s^{(i)} \mid z)], \tag{2}$$

where $p(z)$ is some prior, which we take to be the unit Gaussian, $D_{KL}$ is the Kullback-Leibler divergence, and $\beta$ is a hyperparameter that balances the two terms. The use of $\beta$ values other than one is sometimes referred to as a $\beta$-VAE [15]. The encoder $q_\phi$ parameterizes the mean and log-variance diagonal of a Gaussian distribution, $q_\phi(s) = \mathcal{N}(\mu_\phi(s), \sigma_\phi^2(s))$. The decoder $p_\psi$ parameterizes a Bernoulli distribution for each pixel value. This parameterization corresponds to training the decoder with cross-entropy loss on normalized pixel values. Full details of the hyperparameters are in the Supplementary Material.

## 4   Goal-Conditioned Policies with Unsupervised Representation Learning

To devise a practical algorithm based on goal-conditioned value functions, we must choose a suitable goal representation. In the absence of domain knowledge and instrumentation, a general-purpose choice is to set the goal space $\mathcal{G}$ to be the same as the state observations space $\mathcal{S}$. This choice is fully general as it can be applied to any task, and still permits considerable user control since the user can choose a "goal state" to set a desired goal for a trained goal-conditioned policy. But when the state space $\mathcal{S}$ corresponds to high-dimensional sensory inputs such as images [1] learning a goal-conditioned Q-function and policy becomes exceedingly difficult as we illustrate empirically in Section 5.

Our method jointly addresses a number of problems that arise when working with high-dimensional inputs such as images: sample efficient learning, reward specification, and automated goal-setting. We address these problems by learning a latent embedding using a $\beta$-VAE. We use this latent space to represent the goal and state and retroactively relabel data with latent goals sampled from the VAE prior to improve sample efficiency. We also show that distances in the latent space give us a well-shaped reward function for images. Lastly, we sample from the prior to allow an agent to set and "practice" reaching its own goal, removing the need for humans to specify new goals during training time. We next describe the specific components of our method, and summarize our complete algorithm in Section 4.5.

### 4.1   Sample-Efficient RL with Learned Representations

One challenging problem with end-to-end approaches for visual RL tasks is that the resulting policy needs to learn both perception and control. Rather than operating directly on observations, we embed the state $s_t$ and goals $g$ into a latent space $\mathcal{Z}$ using an encoder $e$ to obtain a latent state $z_t = e(s_t)$ and latent goal $z_g = e(g)$. To learn a representation of the state and goal space, we train a $\beta$-VAE by executing a random policy and collecting state observations, $\{s^{(i)}\}$, and optimize Equation (2). We then use the mean of the encoder as the state encoding, i.e. $z = e(s) \triangleq \mu_\phi(s)$.

After training the VAE, we train a goal-conditioned Q-function $Q(z, a, z_g)$ and corresponding policy $\pi_\theta(z, z_g)$ in this latent space. The policy is trained to reach a goal $z_g$ using the reward function discussed in Section 4.2. For the underlying RL algorithm, we use twin delayed deep deterministic policy gradients (TD3) [13], though any value-based RL algorithm could be used. Note that the policy (and Q-function) operates completely in the latent space. During test time, to reach a specific goal state $g$, we encode the goal $z_g = e(g)$ and input this latent goal to the policy.

As the policy improves, it may visit parts of the state space that the VAE was never trained on, resulting in arbitrary encodings that may not make learning easier. Therefore, in addition to procedure described above, we fine-tune the VAE using both the randomly generated state observations $\{s^{(i)}\}$ and the state observations collected during exploration. We show in Section 8.3 that this additional training helps the performance of the algorithm.

### 4.2   Reward Specification

Training the goal-conditioned value function requires defining a goal-conditioned reward $r(s, g)$. Using Euclidean distances in the space of image pixels provides a poor metric, since similar configurations in the world can be extremely different in image space. In addition to compactly representing high-dimensional observations, we can utilize our representation to obtain a reward function based

on a metric that better reflects the similarity between the state and the goal. One choice for such a reward is to use the negative Mahalanobis distance in the latent space:

$$r(s, g) = -||e(s) - e(g)||_A = -||z - z_g||_A,$$

where the matrix $A$ weights different dimensions in the latent space. This approach has an appealing interpretation when we set $A$ to be the precision matrix of the VAE encoder, $q_\phi$. Since we use a Gaussian encoder, we have that

$$r(s, g) = -||z - z_g||_A \propto \sqrt{\log e_\phi(z_g \mid s)} \tag{3}$$

In other words, minimizing this squared distance in the latent space is equivalent to rewarding reaching states that maximize the probability of the latent goal $z_g$. In practice, we found that setting $A = \mathbf{I}$, corresponding to Euclidean distance, performed better than Mahalanobis distance, though its effect is the same — to bring $z$ close to $z_g$ and maximize the probability of the latent goal $z_g$ given the observation. This interpretation would not be possible when using normal autoencoders since distances are not trained to have any probabilistic meaning. Indeed, we show in Section 5 that using distances in a normal autoencoder representation often does not result in meaningful behavior.

### 4.3 Improving Sample Efficiency with Latent Goal Relabeling

To further enable sample-efficient learning in the real world, we use the VAE to relabel goals. Note that we can optimize Equation (1) using any valid $(s, a, s', g, r)$ tuple. If we could artificially generate these tuples, then we could train our entire RL algorithm without collecting any data. Unfortunately, we do not know the system dynamics, and therefore have to sample transitions $(s, a, s')$ by interacting with the world. However, we have the freedom to relabel the goal and reward synthetically. So if we have a mechanism for generating goals and computing rewards, then given $(s, a, s')$, we can generate a new goal $g$ and new reward $r(s, a, s', g)$ to produce a new tuple $(s, a, s', g, r)$. By artificially generating and recomputing rewards, we can convert a single $(s, a, s')$ transition into potentially infinitely many valid training datums.

For image-based tasks, this procedure would require generating goal images, an onerous task on its own. However, our reinforcement learning algorithm operates directly in the latent space for goals and rewards. So rather than generating goals $g$, we generate latent goals $z_g$ by sampling from the VAE prior $p(z)$. We then recompute rewards using Equation (3). By retroactively relabeling the goals and rewards, we obtain much more data to train our value function. This sampling procedure is made possible by our use of a latent variable model, which is explicitly trained so that sampling from the latent distribution is straightforward.

In practice, the distribution of latents will not exactly match the prior. To mitigate this distribution mismatch, we use a fitted prior when sampling from the prior: we fit a diagonal Gaussian to the latent encodings of the VAE training data, and use this fitted prior in place of the unit Gaussian prior.

Retroactively generating goals is also explored in tabular domains by Kaelbling [18] and in continuous domains by Andrychowicz et al. [2] using hindsight experience replay (HER). However, HER is limited to sampling goals seen along a trajectory, which greatly limits the number and diversity of goals with which one can relabel a given transition. Our final method uses a mixture of the two strategies: half of the goals are generated from the prior and half from goals use the "future" strategy described in Andrychowicz et al. [2]. We show in Section 5 that relabeling the goal with samples from the VAE prior results in significantly better sample-efficiency.

### 4.4 Automated Goal-Generation for Exploration

If we do not know which particular goals will be provided at test time, we would like our RL agent to carry out a self-supervised "practice" phase during training, where the algorithm proposes its own goals, and then practices how to reach them. Since the VAE prior represents a distribution over latent goals and state observations, we again sample from this distribution to obtain plausible goals. After sampling a goal latent from the prior $z_g \sim p(z)$, we give this to our policy $\pi(z, z_g)$ to collect data.

### 4.5 Algorithm Summary

We call the complete algorithm reinforcement learning with imagined goals (RIG) and summarize it in Algorithm 1. We first collect data with a simple exploration policy, though any exploration strategy

**Algorithm 1** RIG: Reinforcement learning with imagined goals

**Require:** VAE encoder $q_\phi$, VAE decoder $p_\psi$, policy $\pi_\theta$, goal-conditioned value function $Q_w$.
1:  Collect $\mathcal{D} = \{s^{(i)}\}$ using exploration policy.
2:  Train $\beta$-VAE on $\mathcal{D}$ by optimizing (2).
3:  Fit prior $p(z)$ to latent encodings $\{\mu_\phi(s^{(i)})\}$.
4:  **for** $n = 0, ..., N - 1$ episodes **do**
5:      Sample latent goal from prior $z_g \sim p(z)$.
6:      Sample initial state $s_0 \sim E$.
7:      **for** $t = 0, ..., H - 1$ steps **do**
8:          Get action $a_t = \pi_\theta(e(s_t), z_g) + $ noise.
9:          Get next state $s_{t+1} \sim p(\cdot \mid s_t, a_t)$.
10:         Store $(s_t, a_t, s_{t+1}, z_g)$ into replay buffer $\mathcal{R}$.
11:         Sample transition $(s, a, s', z_g) \sim \mathcal{R}$.
12:         Encode $z = e(s), z' = e(s')$.
13:         (Probability 0.5) replace $z_g$ with $z'_g \sim p(z)$.
14:         Compute new reward $r = -||z' - z_g||$.
15:         Minimize (1) using $(z, a, z', z_g, r)$.
16:     **end for**
17:     **for** $t = 0, ..., H - 1$ steps **do**
18:         **for** $i = 0, ..., k - 1$ steps **do**
19:             Sample future state $s_{h_i}, t < h_i \leq H - 1$.
20:             Store $(s_t, a_t, s_{t+1}, e(s_{h_i}))$ into $\mathcal{R}$.
21:         **end for**
22:     **end for**
23:     Fine-tune $\beta$-VAE every $K$ episodes on mixture of $\mathcal{D}$ and $\mathcal{R}$.
24: **end for**

could be used for this stage, including off-the-shelf exploration bonuses [29, 4] or unsupervised reinforcement learning methods [9, 12]. Then, we train a VAE latent variable model on state observations and finetune it over the course of training. We use this latent variable model for multiple purposes: We sample a latent goal $z_g$ from the model and condition the policy on this goal. We embed all states and goals using the model's encoder. When we train our goal-conditioned value function, we resample goals from the prior and compute rewards in the latent space using Equation (3). Any RL algorithm that trains Q-functions could be used, and we use TD3 [13] in our implementation.

# 5 Experiments

Our experiments address the following questions:

1. How does our method compare to prior model-free RL algorithms in terms of sample efficiency and performance, when learning continuous control tasks from images?
2. How critical is each component of our algorithm for efficient learning?
3. Does our method work on tasks where the state space cannot be easily specified ahead of time, such as tasks that require interaction with variable numbers of objects?
4. Can our method scale to real world vision-based robotic control tasks?

For the first two questions, we evaluate our method against a number of prior algorithms and ablated versions of our approach on a suite of the following simulated tasks. *Visual Reacher*: a MuJoCo [47] environment with a 7-dof Sawyer arm reaching goal positions. The arm is shown the left of Figure 2. The end-effector (EE) is constrained to a 2-dimensional rectangle parallel to a table. The action controls EE velocity within a maximum velocity. *Visual Pusher*: a MuJoCo environment with a 7-dof Sawyer arm and a small puck on a table that the arm must push to a target push. *Visual Multi-Object Pusher*: a copy of the Visual Pusher environment with two pucks. *Visual Door*: a Sawyer arm with a door it can attempt to open by latching onto the handle. *Visual Pick and Place*: a Sawyer arm with a small ball and an additional dimension of control for opening and closing the gripper. Detailed descriptions of the environments are provided in the Supplementary Material.

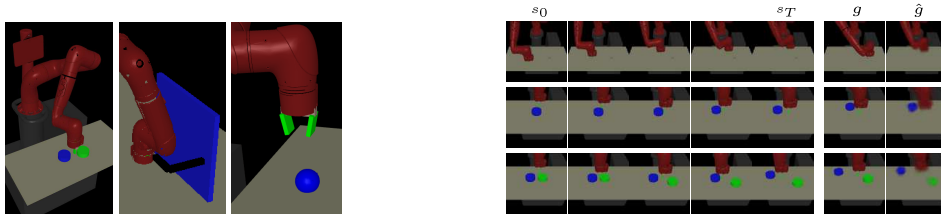

Figure 2: (Left) The simulated pusher, door opening, and pick-and-place environments are pictured. (Right) Test rollouts from our learned policy on the three pushing environments. Each row is one rollout. The right two columns show a goal image $g$ and its VAE reconstruction $\hat{g}$. The images to their left show frames from a trajectory to reach the given goal.

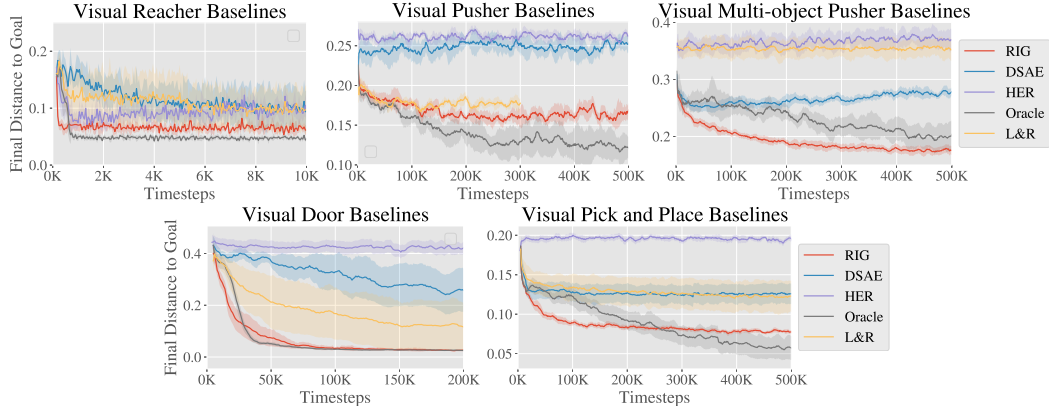

Figure 3: Simulation results, final distance to goal vs simulation steps[2]. RIG (red) consistently outperforms the baselines, except for the oracle which uses ground truth object state for observations and rewards. On the hardest tasks, only our method and the oracle discover viable solutions.

Solving these tasks directly from images poses a challenge since the controller must learn both perception and control. The evaluation metric is the distance of objects (including the arm) to their respective goals. To evaluate our policy, we set the environment to a sampled goal position, capture an image, and encode the image to use as the goal. Although we use the ground-truth positions for evaluation, **we do not use the ground-truth positions for training the policies**. The only inputs from the environment that our algorithm receives are the image observations. For Visual Reacher, we pretrained the VAE with 100 images. For other tasks, we used 10,000 images.

We compare our method with the following prior works. *L&R*: Lange and Riedmiller [20] trains an autoencoder to handle images. *DSAE*: Deep spatial autoencoders [11] learns a spatial autoencoder and uses guided policy search [23] to achieve a single goal image. *HER*: Hindsight experience replay [2] utilizes a sparse reward signal and relabeling trajectories with achieved goals. *Oracle*: RL with direct access to state information for observations and rewards.

To our knowledge, no prior work demonstrates policies that can reach a variety of goal images without access to a true-state reward function, and so we needed to make modifications to make the comparisons feasible. L&R assumes a reward function from the environment. Since we have no state-based reward function, we specify the reward function as distance in the autoencoder latent space. HER does not embed inputs into a latent space but instead operates directly on the input, so we use pixel-wise mean squared error (MSE) as the metric. DSAE is trained only for a single goal, so we allow the method to generalize to a variety of test goal images by using a goal-conditioned Q-function. To make the implementations comparable, we use the same off-policy algorithm, TD3 [13], to train L&R, HER, and our method. Unlike our method, prior methods do not specify how to select goals during training, so we favorably give them real images as goals for rollouts, sampled from the same distribution that we use to test.

We see in Figure 3 that our method can efficiently learn policies from visual inputs to perform simulated reaching and pushing, without access to the object state. Our approach substantially outperforms the prior methods, for which the use of image goals and observations poses a major challenge. HER struggles because pixel-wise MSE is hard to optimize. Our latent-space rewards are much better shaped and allow us to learn more complex tasks. Finally, our method is close to the state-based "oracle" method in terms of sample efficiency and performance, without having any access to object state. Notably, in the multi-object environment, our method actually outperforms the oracle, likely because the state-based reward contains local minima. Overall, these result show that our method is capable of handling raw image observations much more effectively than previously proposed goal-conditioned RL methods. Next,

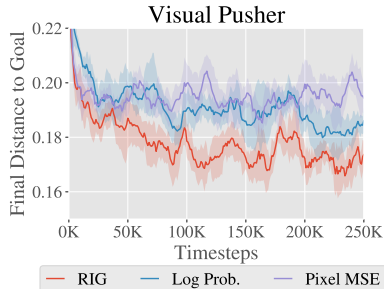

Figure 4: Reward type ablation results. RIG (red), which uses latent Euclidean distance, outperforms the other methods.

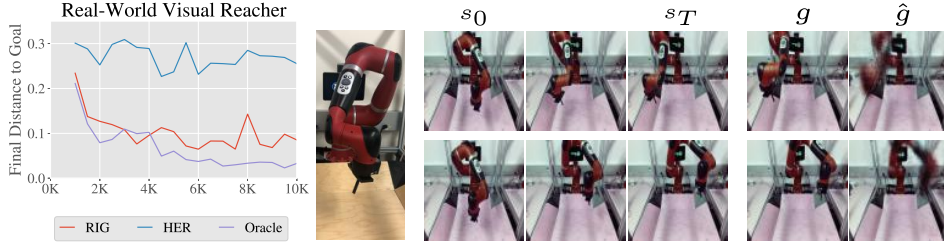

Figure 7: (Left) Our method compared to the HER baseline and oracle on a real-world visual reaching task. (Middle) Our robot setup is pictured. (Right) Test rollouts of our learned policy.

we perform ablations to evaluate our contributions in isolation. Results on Visual Pusher are shown but see the Supplementary Material (section 8) for experiments on all three simulated environments.

**Reward Specification Comparison** We evaluate how effective distance in the VAE latent space is for the Visual Pusher task. We keep our method the same, and only change the reward function that we use to train the goal-conditioned valued function. We include the following methods for comparison: *Latent Distance*, which uses the reward used in RIG, i.e. $A = \mathbf{I}$ in Equation (3); *Log Probability*, which uses the Mahalanobis distance in Equation (3), where $A$ is the precision matrix of the encoder; and *Pixel MSE*, which uses mean-squared error (MSE) between state and goal in pixel space. [3] In Figure 4, we see that latent distance significantly outperforms log probability. We suspect that small variances of the VAE encoder results in drastically large rewards, making the learning more difficult. We also see that latent distances results in faster learning when compared to pixel MSE.

**Relabeling Strategy Comparison** As described in section 4.3, our method uses a novel goal relabeling method based on sampling from the generative model. To isolate how much our new goal relabeling method contributes to our algorithm, we vary the resampling strategy while fixing other components of our algorithm. The resampling strategies that we consider are: *Future*, relabeling the goal for a transition by sampling uniformly from future states in the trajectory as done in Andrychowicz et al. [2]; *VAE*, sampling goals from the VAE only; *RIG*, relabeling goals with probability $0.5$ from the VAE and probability $0.5$ using the future strategy; and *None*, no relabeling. In Figure 5, we see that sampling from the VAE and Future is significantly better than not relabeling at all. In RIG, we use an equal mixture of the VAE and Future sampling strategies, which performs best by a large margin. Appendix section 8.1 contains results on all simulated environments, and section 8.4 considers relabeling strategies with a known goal distribution.

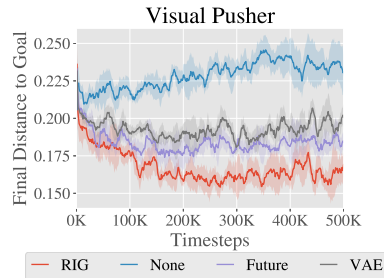

Figure 5: Relabeling ablation.

**Learning with Variable Numbers of Objects** A major advantage of working directly from pixels is that the policy input can easily represent combinatorial structure in the environment, which would be difficult to encode into a fixed-length state vector even if a perfect perception system were available. For example, if a robot has to interact with different combinations and numbers of objects, picking a single MDP state representation would be challenging, even with access to object poses. By directly processing images for both the state and the goal, no modification is needed to handle the combinatorial structure: the number of pixels always remains the same, regardless of how many objects are in the scene.

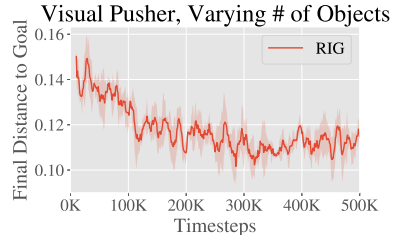

Figure 6: Training curve for learning with varying number of objects.

We demonstrate that our method can handle this difficult scenario by evaluating on a task where the environment, based on the Visual Multi-Object Pusher, randomly contains zero, one, or two objects in each episode during testing. During training, each episode still always starts with both objects in the scene, so the experiments tests whether a trained policy can handle variable numbers of objects at test time. Figure 6 shows that our method can learn to solve this task successfully, without decrease

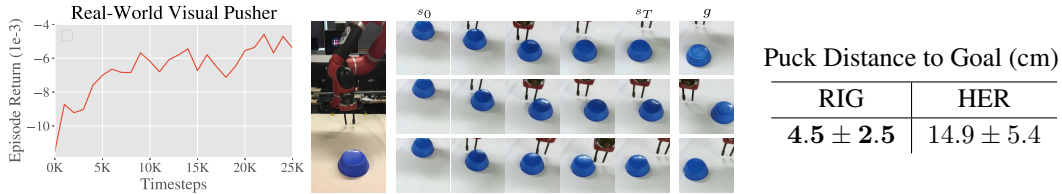

| | Puck Distance to Goal (cm) | |
|---|---|---|
| | RIG | HER |
| | $\mathbf{4.5 \pm 2.5}$ | $14.9 \pm 5.4$ |

Figure 8: (Left) The learning curve for real-world pushing. (Middle) Our robot pushing setup is pictured, with frames from test rollouts of our learned policy. (Right) Our method compared to the HER baseline on the real-world visual pushing task. We evaluated the performance of each method by manually measuring the distance between the goal position of the puck and final position of the puck for 15 test rollouts, reporting mean and standard deviation.

in performance from the base setting where both objects are present (in Figure 3). Developing and demonstrating algorithms that solve tasks with varied underlying structure is an important step toward creating autonomous agents that can handle the diversity of tasks present "in the wild."

## 5.1 Visual RL with Physical Robots

RIG is a practical and straightforward algorithm to apply to real physical systems: the efficiency of off-policy learning with goal relabeling makes training times manageable, while the use of image-based rewards through the learned representation frees us from the burden of manually design reward functions, which itself can require hand-engineered perception systems [39]. We trained policies for visual reaching and pushing on a real-world Sawyer robotic arm, shown in Figure 7. The control setup matches Visual Reacher and Visual Pusher respectively, meaning that **the only input from the environment consists of camera images**.

We see in Figure 7 that our method is applicable to real-world robotic tasks, almost matching the state-based oracle method and far exceeding the baseline method on the reaching task. Our method needs just 10,000 samples or about an hour of real-world interaction time to solve visual reaching.

Real-world pushing results are shown in Figure 8. To solve visual pusher, which is more visually complicated and requires reasoning about the contact between the arm and object, our method requires about 25,000 samples, which is still a reasonable amount of real-world training time. Note that unlike previous results, we do not have access to the true puck position during training so for the learning curve we report test episode returns on the VAE latent distance reward. We see RIG making steady progress at optimizing the latent distance as learning proceeds.

## 6 Discussion and Future Work

In this paper, we present a new RL algorithm that can efficiently solve goal-conditioned, vision-based tasks without access to any ground truth state or reward functions. Our method trains a generative model that is used for multiple purposes: we embed the state and goals using the encoder; we sample from the prior to generate goals for exploration; we also sample latents to retroactively relabel goals and rewards; and we use distances in the latent space for rewards to train a goal-conditioned value function. We show that these components culminate in a sample efficient algorithm that works directly from vision. As a result, we are able to apply our method to a variety of simulated visual tasks, including a variable-object task that cannot be easily represented with a fixed length vector, as well as real world robotic tasks. Algorithms that can learn in the real world and directly use raw images can allow a single policy to solve a large and diverse set of tasks, even when these tasks require distinct internal representations.

## 7 Acknowledgements

We would like to thank Aravind Srinivas and Pulkit Agrawal for useful discussions, and Alex Lee for helpful feedback on an initial draft of the paper. We would also like to thank Carlos Florensa for making multiple useful suggestions in later version of the draft. This work was supported by the National Science Foundation IIS-1651843 and IIS-1614653, a Huawei Fellowship, Berkeley DeepDrive, Siemens, and support from NVIDIA.

## Footnotes

[1]We make the simplifying assumption that the system is Markovian with respect to the sensory input, and one could incorporate memory into the state for partially observed tasks.

[2]In all our simulation results, each plot shows a 95% confidence interval of the mean across 5 seeds.

[3]To compute the pixel MSE for a sampled latent goal, we decode the goal latent using the VAE decoder, $p_\psi$, to generate the corresponding goal image.

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
