[Supplementary Material]

# Supplementary Material

## 8 Complete Ablative Results

### 8.1 Relabeling strategy ablation

In this experiment, we compare different goal resampling strategies for training the Q function. We consider: *Future*, relabeling the goal for a transition by sampling uniformly from future states in the trajectory as done in Andrychowicz et al. [2]; *VAE*, sampling goals from the VAE only; *RIG*, relabeling goals with probability $0.5$ from the VAE and probability $0.5$ using the future strategy; and *None*, no relabeling. Figure 9 shows the effect of different relabeling strategies with our method.

Figure 9: Relabeling ablation simulated results, showing final distance to goal vs environment steps. RIG (red), which uses a mixture of VAE and future, consistently matches or outperforms the other methods.

### 8.2 Reward type ablation

In this experiment, we change only the reward function that we use to train the goal-conditioned valued function to show the effect of using the latent distance reward. We include the following methods for comparison: *Latent Distance*, which is the reward used in RIG, i.e. $A = \mathbf{I}$ in Equation (3); *Log Probability*, which uses the Mahalanobis distance in Equation (3), where $A$ is the precision matrix of the encoder; and *Pixel MSE*, which computes mean-squared error (MSE) between state and goal in pixel space. To compute the pixel MSE for a sampled latent goal, we decode the goal latent using the VAE decoder, $p_\psi$, to generate the corresponding goal image. Figure 10 shows the effect of different rewards with our method.

Figure 10: Reward type ablation simulated results, showing final distance to goal vs environment steps. RIG (red), which uses latent distance for the reward, consistently matches or outperforms the other reward types.

### 8.3 Online training ablation

Rather than pre-training the VAE on a set of images collected by a random policy, here we train the VAE in an online manner: the VAE is not trained when we initially collect data with our policy. After every 3000 environment steps, we train the VAE on all of the images observed by the policy. We show in Figure 11 that this online training results in a good policy and is substantially better than leaving the VAE untrained. These results show that the representation learning can be done simultaneously as the reinforcement learning portion of RIG, eliminating the need to have a predefined set of images to train the VAE.

The Visual Pusher experiment for this ablation is performed on a slightly easier version of the Visual Pusher used for the main results. In particular, the goal space is reduced to be three quarters of its original size in the lateral dimension.

Figure 11: Online vs offline VAE training ablation simulated results, showing final distance to goal vs environment steps. Given no pre-training phase, training the VAE online (red), outperforms no training of the VAE, and also performs well.

## 8.4 Comparison to Hindsight Experience Replay

In this section, we study in isolation the effect of sampling goals from the goal space directly for Q-learning, as covered in Section 4.3. Like hindsight experience replay [2], in this section we assume access to state information and the goal space, so we do not use a VAE.

To match the original work as closely as possible, this comparison was based off of the OpenAI baselines code [34] and we compare on the same Fetch robotics tasks. To minimize sample complexity and due to computational constraints, we use single threaded training with `rollout_batch_size=1`, `n_cycles=1`, `batch_size=256`. For testing, `n_test_rollouts=1` and the results are averaged over the last 100 test episodes. Number of updates per cycle corresponds to `n_batches`.

On the plots, "Future" indicates the future strategy as presented in Andrychowicz et al. [2] with $k = 4$. "Ours" indicates resampling goals with probability 0.5 from the "future" strategy with $k = 4$ and probability 0.5 uniformly from the environment goal space. Each method is shown with dense and sparse rewards.

Figure 12: Comparison between our relabeling strategy and HER. Each column shows a different task from the OpenAI Fetch robotics suite. The top row uses 64 gradient updates per training cycle and the bottom row uses 256 updates per cycle. Our relabeling strategy is significantly better for both sparse and dense rewards, and for higher number of updates per cycle.

Results are shown in Figure 12. Our resampling strategy with sparse rewards consistently performs the best on the three tasks. Furthermore, it performs reasonably well with dense rewards, unlike HER alone which often fails with dense rewards. While the evaluation metric used here, success rate, is favorable to the sparse reward setting, learning with dense rewards is usually more sample efficient on most tasks and being able to do off-policy goal relabeling with dense rewards is important for RIG.

Finally, as the number of gradient updates per training cycle is increased, the performance of our strategy improves while HER does not improve and sometimes performs worse. As we apply

reinforcement learning to real-world tasks, being able to reduce the required number of samples on hardware is one of the key bottlenecks. Increasing the number of gradient updates costs more compute but reduces the number of samples required to learn the tasks.

## 9   Hyperparameters

Table 1 lists the hyperparameters used for the experiments.

| Hyperparameter | Value | Comments |
|---|---|---|
| Mixture coefficient $\lambda$ | 0.5 | See relabeling strategy ablation |
| # training batches per time step | 4 | Marginal improvements after $4$ |
| Exploration Policy | OU, $\theta = 0.15, \sigma = 0.3$ | Outperformed Gaussian and $\epsilon$-greedy |
| $\beta$ for $\beta$-VAE | 5 | Values around $[1, 10]$ were effective |
| Critic Learning Rate | $10^{-3}$ | Did not tune |
| Critic Regularization | None | Did not tune |
| Actor Learning Rate | $10^{-3}$ | Did not tune |
| Actor Regularization | None | Did not tune |
| Optimizer | Adam | Did not tune |
| Target Update Rate ($\tau$) | $10^{-2}$ | Did not tune |
| Target Update Period | 2 time steps | Did not tune |
| Target Policy Noise | 0.2 | Did not tune |
| Target Policy Noise Clip | 0.5 | Did not tune |
| Batch Size | 128 | Did not tune |
| Discount Factor | 0.99 | Did not tune |
| Reward Scaling | $10^{-4}$ | Did not tune |
| Normalized Observations | False | Did not tune |
| Gradient Clipping | False | Did not tune |

Table 1: Hyper-parameters used for all experiments.

## 10   Environment Details

Below we provide a more detailed description of the simulated environments.

*Visual Reacher*: A MuJoCo environment with a 7-DoF Sawyer arm reaching goal positions. The arm is shown on the left of Figure 2 with two extra objects for the Visual Multi-Object Pusher environment (see below). The end-effector (EE) is constrained to a 2-dimensional rectangle parallel to a table. The action controls EE velocity within a maximum velocity. The underlying state is the EE position $e$, and the underlying goal is to reach a desired EE position, $g_e$.

*Visual Pusher*: A MuJoCo environment with a 7-DoF Sawyer arm and a small puck on a table that the arm must push to a target position. Control is the same as in Visual Reacher. The underlying state is the EE position, $e$ and puck position $p$. The underlying goal is for the EE to reach a desired position $g_e$ and the puck to reach a desired position $p$.

*Visual Multi-Object Pusher*: A copy of the Visual Pusher environment with two pucks. The underlying state is the EE position, $e$ and puck positions $p_1$ and $p_2$. The underlying goal is for the EE to reach desired position $g_e$ and the pucks to reach desired positions $g_1$ and $g_2$ in their respective halves of the workspace. Each puck and respective goal is initialized in half of the workspace.

Videos of our method in simulated and real-world environments can be found at `https://sites.google.com/site/visualrlwithimaginedgoals/`.