[Reviews · NeurIPS 2018]

Reviewer 1



This paper proposes an algorithm for learning goal-conditioned RL policy, in which a goal is defined as a single image. The authors propose to encode a state (an image) to a vector in latent space using variational autoencoder, and define reward functions inside the latent space. The paper shows that such reward function outperforms baseline such as pixel based reward functions. The authors then proposed latent goal relabeling, which generates new goals and rewards given an exist tuple (s, a, s’). In this way, the off-policy algorithm essentially obtains more training data. Finally, the authors propose goal imagination, which samples goals from latent space during training, essentially allowing training without specifying a particular goal. (interacting with the environment is still requires, because the transition function is unknown). The experiment results justified the proposed methods, showing that the proposed algorithm is more sample efficient and achieve better performance. The proposed method is novel and the experiments are thorough, and I'm inclined to accept this paper. Here are some suggestions/comments: 1) A goal in the paper is defined as a single image, which limits the application of the algorithm. Usually goals are more abstract than an image (e.g. in auto driving, probably one goal is to stay in the lane), or sometimes multiple images maps to the same goal. Indeed, defining the goal space the same as observation space is convenient, but has its own limitations. 2) The algorithm first trains a VAE model using some exploration policy, and then fixes the VAE parameters and trains the policy. However, this requires the exploration policy to explore extensively to cover better the state space so as to collect enough data to train the VAE model. This is a time and sample consuming step. Is it possible to incrementally update both VAE and policy parameters? 3) Experiments of reward specification comparison (Figure 4). It would be nice if the authors could add another baseline which is GRiLL plus hand-specified reward function (the reward function Oracle used in Figure 3). It would also be nice if the authors can add another figure which demonstrates the correlation between the proposed reward function and the hand-specified reward function used by Oracle. 4) Line 291: the authors state “VAE is significantly better than HER or not relabeling at all”. However, Figure 5 shows that VAE (the grey line) is doing worse than HER (purple line). The labels in the figure is wrong? 5) Figure 6: it would be better if the authors also run baselines for the variable numbers of objects setting.

Reviewer 2



The paper introduces an approach for learning general purpose skills by combining learning a latent representation of images with goal conditioned policies. A visual representation of raw observations is learnt using a VAE model that can also be used to generate arbitrary goals for a goal conditioned policy. The resulting learnt policy can operate with raw image observations in physical control tasks. The framework can also be interpreated as a generalization of Hindisght Experience Replay for control tasks. A latent variable model is used to learnt a representation of raw image inputs based on VAE framework. The authors motivate that such a model not only learns a good representation of raw inputs, but the prior on the latent variables can also be used to generate random goals for learning a goal conditioned policy. The proposed latent variable model is used for several objectives, specifically to compute arbitrary reward functions based on distances in the latent space between the current state and randomly generated goal. The overall RL objective is trained alongside with a \beta-VAE objective._x000b__x000b_Even though the paper is trying to motivate an interesting latent variable model that can handle raw images for control tasks, there are several aspects of the paper which make the overall research contribution complex. The paper proposes several objectives, and it is not clear why the proposed approach exactly works better. I also do not think the proposed method of learning a general purpose goal conditioned policy, as motivated in the paper, is indeed what it claims to be. _x000b_The experimental results demonstrated are not fully satisfying. Even if the authors compare to several baselines including HER, they mention modifications being made to the baseline algorithms. It would be been useful to see how the proposed method works on other tasks, instead of simply on the Reacher and Pusher baselines, along with a comparision with the original baselines (e.g HER uses similar experiment setups with Pusher and Reacher). The range of experimental results is very limited, and the modifications made to the baselines raises questions about reproducibility of these results. _x000b__x000b_Another confusing aspect is the use of TD3 - why is TD3 used in the first place instead of standard off-policy RL algorithms for control tasks? Was there a specific reason to use TD3?_x000b__x000b_Overall, the paper introduces an interesting idea to use a latent variable model to achieve several objectives, but fails to sufficiently demonstrate the proposed approach experimentally on a range of tasks. The limited set of experimental results are not sufficiently convincing, and it is not exactly clear how the proposed approach can be implemented or can be made to work in practice. The paper is trying to motivate an interesting idea, but may not be useful for practitioners in the field for how to use the proposed approach. More detailed discussion of the algorithm and implementation details are required. _x000b__x000b_I suggest the authors should make it more clear how the experiments with the proposed method can be implemented and reproduced in practice._x000b_ Quality : Good. Clarity : Average. The paper is at times difficult to follow as it tries to motivate the proposed approach from several perspectives, but does not provide enough intuitions or experimental demonstration of how they are achieved. _x000b_ Originality : Slightly above average_x000b__x000b_Significance : Average. Overall Rating : Marginally above acceptance level. I think the paper is introducing an interesting approach for raw image based control tasks, but does not still provide sufficient experimental results to demonstrate it. However, I think the overall research contribution is novel and maybe considered for acceptance.

Reviewer 3



Standard RL framework often focuses on learning policies that are specific to individual tasks with hand-specified reward functions. This paper proposes an RL framework that jointly learns representations of inputs and policies by practicing to reach self-specified random goals during training. It incorporates unsupervised representation learning into sample-efficient off-policy goal-conditioned reinforcement learning. The proposed model is extensively evaluated on vision-based tasks without access to any ground truth state or reward functions, successfully demonstrating the advantages of the proposed model. The paper is self-contained, well written and easy to read. The proposed methods simply trains a β-VAE over raw sensory inputs and used for 1) to embed the state and goals using the encoder, 2) to sample goals for exploration from the prior, 3) to sample latent representations to retroactively relabel goals and rewards and 4) to estimate distances in the latent space for rewards to train a goal-conditioned value function. Despite its simplicity, models seem to be very effective in various ways: it provides a more structured representation of complex inputs such as images for RL; it allows for the sampling of new states, which can be used to set synthetic goals; it allows relabelling the goals and rewards facilitates the training of the value function with data augmentation; and it provides a well-shaped reward functions over latent representation then pixel-wise Euclidean distance for images. Evaluation is another strength of the paper. The model is compared against prior model-free RL algorithms over standard benchmarks. The paper also reports on ablation studies and scaling issues. Line 291-292: "In Figure 5, we see that sampling from the VAE is significantly better than HER or not relabeling at all. Lastly, a mixture of the the VAE and HER sampling performs the best, which we use in GRiLL." Should VAE and HER be swapped? In figure 5, HER seems to be performing better than VAE. Also, restate your intuitions behind why GriLL performs the best here. This should be considered at other places in the Experiment section. Typos and Other comments: Figure 3 Caption: Footnote 4 missing. Line 102: E[] Remove spaces before footnote pointers. Footnotes 3 and 4 are missing.